# Use of LoopDeelab during the COVID-19 Pandemic: An Innovative Device for Field Diagnosis

**DOI:** 10.3390/v14092062

**Published:** 2022-09-17

**Authors:** Nefert Candace Dossou, Isidore Gaubert, Elodie Maille, Remy Morello, Renaud Cassier, Cécile Schanen, Jean-Jacques Dutheil, Louis-Marie Rocque, Astrid Vabret, Meriadeg Ar Gouilh

**Affiliations:** 1Virology Department, Caen University Hospital, 14033 Caen, France; 2INSERM U1311 DynaMicURe, Normandy University, UNICAEN, UNIROUEN, 14033 Caen, France; 3LoopdeeScience, 14000 Caen, France; 4Biostatistics and Clinical Research Unit, Caen University Hospital, 14033 Caen, France; 5Department of Clinical Research and Innovation, Caen University Hospital, 14033 Caen, France

**Keywords:** RAPID COVID, Loopdeetect, Loopdeelab, COVID-19, SARS-CoV-2, LAMP PCR

## Abstract

Rapid and accurate diagnosis of SARS-CoV-2 infection is essential for the management of the COVID-19 outbreak. RT-LAMP LoopDeetect COVID-19 (LoopDeescience, France) is a rapid molecular diagnostic tool which operates with the LoopDeelab (LoopDeescience, France) device. RAPID COVID is a prospective double-blind research protocol which was conducted to evaluate the concordance between Loopdeetect COVID-19 and RT-PCR Allplex 2019 n-Cov (Seegene, Korea). Between 11 May 2020 and 14 June 2021, a total of 1122 nasopharyngeal swab specimens were collected, of which 741 were finally analysed. There were 32 “positive” and “indeterminate” RT-PCR results. The intrinsic performances of Loopdeetect COVID-19 are equivalent to other commercial RT-LAMP PCR COVID-19 kits, with a sensitivity and specificity of 69.23% [CI 95%: 48.21–85.67] and 100% [CI 95%: 99.58–100.00], respectively. To the best of our knowledge, LoopDeelab is the only LAMP PCR diagnostic device allowing such a fast and reliable analysis with low-cost equipment; this makes it a new and innovative technology, designed for field use. This device being portable, the development of other detection kits will be useful for the management of epidemics with a high attack rate and would facilitate the rapid application of health measures.

## 1. Introduction

In late December 2019, a new disease named COVID-19 (Coronavirus Infectious Disease 2019), due to a new *Sarbecovirus*, *Betacoronavirus*, named SARS-CoV-2, was notified in Wuhan, China. The World Health Organization (WHO) declared “A public health emergency of international concern” on 30 January 2020. More than two years later, the virus has spread to the whole world and has caused more than 540,000,000 cases, including more than 6,320,000 deaths (with a large majority in USA, Brazil and India—23 June 2022, WHO) [1]. In this context, the implementation of rapid tests, effective in sensitivity and specificity, for diagnosis of SARS-CoV2 infection, was a priority announced by the WHO and the French national research agency.

Nasopharyngeal swabs collected in a liquid universal transport media and combined with standardised nucleic-acid extraction and real-time RT-PCR (RT-qPCR) is the gold standard method to detect SARS-CoV-2. These different steps necessitate from 4 to 8 h of handling and incubation time, which in turn induces a sample-to-result time ranging from 6 to 24 h in diagnostic labs. These time constraints, processing delays and the necessity of expert technicians were a major impediment to the reactivity and efficiency of diagnostic laboratories and to testing policies. This is why, throughout the COVID-19 pandemic, rapid diagnostic tests have been developed, such as antigenic tests, rapid PCR or RT-LAMP PCR.

RT-LAMP (Reverse-Transcription and Loop-mediated isothermal AMPlification) might advantageously increase testing efficiency, as it allows all-in-one rapid testing, including RNA reverse transcription and DNA amplification, without a requirement for temperature cycles. LAMP does not even require complex devices and laboratory infrastructure; RT-LAMP can use low-cost and mobile equipment already available in most molecular diagnostic laboratories and utilises the inhibitor-tolerant Bst polymerase [2]. These traits make LAMP useful in field detection assays [3,4,5,6,7]. The use of LAMP in detection systems has been explored for numerous micro-organisms and epidemic pathogens (Zika-virus, Ebolavirus, Malaria) [8], several of which are validated by the WHO [9] and by the Food and Agriculture Organization (FAO) [3,10]. For COVID-19 diagnosis, several commercial RT-LAMP PCR tests are already available, but their intrinsic performances are quite low compared to RT-PCR tests and seem heterogeneous, with sensitivities ranging from 48.0% to 71.7% [11,12,13,14].

RT-LAMP LoopDeetect COVID-19 (LoopDeescience, Caen, France) is an innovative rapid molecular diagnostic tool for SARS-CoV-2 infection. It operates with the LoopDeelab (LoopDeescience, Caen, France) mobile device, whose dimension (6 × 6 × 6 cm) and low energy requirements (USB-c connection) allow a simple point-of-care analysis.

RAPID COVID is a prospective double-blind non-interventional research protocol. It was conducted to evaluate the concordance between the rapid molecular detection tool (LoopDeetect COVID-19) results and the routine molecular diagnosis (RT-PCR Allplex 2019 n-Cov assay kit, Seegene, Korea) results at the virology department of Caen University Hospital. The secondary aim of this protocol was to assess the feasibility of rapid screening in the midst of a pandemic.

## 2. Materials and Methods

RAPID COVID is a prospective double-blind non-interventional research protocol. In accordance with the protocol, the study participants were prospectively enrolled at two different sites: a private external testing site and at the COVID-19 screening ceffnter of CAEN University Hospital. All samples were submitted to routine molecular diagnosis (RT-PCR Allplex 2019 n-Cov assay kit, Seegene, Korea) at the Virology department of CAEN University Hospital and to RT-LAMP assay (LoopDeetect COVID-19) performed on testing sites.

### 2.1. Ethical

This protocol received ethical approval from the Comité de Protection des Personnes (CPP) Sud Méditerrannée V on 20 April 2020 under the number 2020-A01007-32/SI:20.04.10.37320. It was classified as research involving human persons of third category and complied with the “methodologie de reference” MR003 from the Commission Nationale Informatique et Libertés (CNIL). Three unblindings were planned after the inclusion of respectively 100, 300 and 1000 patients. Two substantial modifications to this protocol received ethical approval from the CPP on 17 August 2020 and on 6 April 2021.

Each patient included was previously informed about the protocol and their non-objection was obtained from one of the investigators.

All patient data has been anonymised. They were managed in a database administrated by the promoter (Ennov Clinical software, Ennov Group, Paris, France,). Access to this database was limited to authorised persons only (promoter team, investigators). User authentication was done with custom username and password. All database connections were recorded in the connection history.

### 2.2. Participants

The inclusion criteria were: major and french-speaking patients with an affiliation to the French social security system. The exclusion criteria were: Hemophilia, repeated epistaxis, impossibility of performing the nasopharyngeal swab, being under guardianship or curatorship or placed under protective measures, and patient refusal to participate.

The investigators explained to each participant the study, its objectives, and obtained the oral consent of the patient, as required by the Ethical committee. A trained agent carried out a questionnaire exploring age, sex, size and body weight, reasons for testing, notion of contact case and of COVID-19-like-symptoms. Patients were considered symptomatics if they had any of the following symptoms: hyperthermia, myalgia, cough, chest pain, dyspnea, rhinitis, headache, diarrhea, anosmia, ageusia, rash. An anonymous identification number was assigned to each patient enrolled.

All information necessary for registering patients on the SI-DEP database was collected. SI-DEP is a french screening information system that automatically stores all the results of COVID-19 tests carried out by more than 600 public and private analysis laboratories. It allows contact tracing and simplifies the management of people tested positively by health insurance.

All study participants received only the routine molecular diagnosis (RT-PCR) results via SI-DEP within the usual timeframe. All RT-PCR results has been validated by competent medical biologists who could had a comment to the result if they deemed it necessary or request a control of the result by another PCR assay. Within the framework of this study, only the results obtained with Allplex 2019 n-Cov assay kit (Seegene, Korea) will be analysed.

### 2.3. Testing Sites

The study participants were prospectively enrolled at a private external testing site in prefabricated premises. On 17 August 2020, the protocol has been updated, i.e. the testing site was changed for the COVID-19 screening center of Caen University Hospital. (Figure 1), which is a free screening center open to everyone and every morning of the week, by appointment.

### 2.4. Sampling

Nasopharyngeal swabs were performed as paired samples (1 swab per nostril per patient) by previously trained and authorised personnel using two flocked swabs (IMPROSWAB, Improve, China). One of the swabs was stirred in a universal liquid transport media following the Standard Operational Procedures (S.O.Ps) of the Virology department of Caen University hospital for routine molecular diagnosis. The other swab was stirred in a 0.9% NaCl medium following the manufacturer’s recommendations and then analysed onsite using the LoopDeetect method. Both methods were carried out double-blind.

On 6 April 2021, the protocol was updated, i.e., the sampling method was modified as following: Only one swab was used for both nostrils per patient and then stirred in a 2 mL 0.9% NaCl medium. The resulting sample was split in two aliquots: 1 mL was sent to the Virology department of Caen University hospital for routine molecular diagnosis and 1 mL was analysed directly onsite using the LoopDeetect method (Figure 1).

The remaining volume of each sample or nucleic acids was divided into aliquots and stored at −80 °C.

### 2.5. LoopDeetect Method

LoopDeetect COVID-19 (LoopDeescience, Caen, France) is a LAMP PCR method that does not require nucleic acid extraction. A slightly variable region of the *Betacoronavirus* genome (RdRp gene) is amplified with 3 sets of primers. RT-LAMP assays were carried out by previously trained and authorised personnel. A training certificate was delivered by LoopDeescience for each authorised person.

RT-LAMP assay (LoopDeetect COVID-19) were performed on a LoopDeelab (Loopdeescience, Caen, France) device for 45 min at 63 °C (Figure 2 and Figure 3).

### 2.6. Routine Molecular Diagnosis at Virology Department of Caen University Hospital

All samples were analysed with a commercial multiplex real time RT-qPCR targeting the envelope gene (E) of the Sarbecovirus, RNA-dependent RNA polymerase (RdRp) and nucleocapsid (N) viral genes (Allplex 2019 n-Cov assay kit, Seegene, Korea). RT-qPCR assays were performed on CFX (Bio-rad Laboratories, Inc., Hercules, CA, USA). The PCR programme comprised two different steps. Step 1 ran 1 cycle at 50 °C for 20 min and 1 cycle at 95 °C for 15 min. Step 2 presented 45 cycles of 94 °C for 15 s and 58 °C for 38 s.

Viral nucleic acids were extracted using the Nucleospin RNA kit (Macherey-Nagel, Düren, Germany) on Ideal 32 (ID Solutions). The final elution was 100 µL.

The results were analysed using the Seegene Viewer (Seegene, Korea) according to the manufacturer’s instructions. Valid results were defined as amplification of the internal control gene with a Ct < 35. A sample was considered positive if at least two of the RdRp, N or E genes were amplified with a Ct < 40. Negative samples were defined as no amplification of any viral genes. Indeterminate samples were defined as amplification of only one of the RdRp, N or E genes with a Ct < 40.

### 2.7. Statistical Analysis

The gold standard was the result of routine molecular diagnosis (RT-PCR Allplex 2019 n-Cov assay kit, Seegene, Korea). Sensitivity, specificity, negative predictive values and positive predictive values were computed. For the 95% confidence intervals, the exact method was used. We considered the RdRp gene—which is specific for SARS-CoV-2 and is the common target of the two compared methods—to calculate different Ct categories. We used the Student’s t test or Fisher’s exact test to compare different variables between symptomatic and asymptomatic groups. Levene’s test was used to check if our data satisfied the homogeneity of variance assumption before performing the Student’s t test. All statistical analyses was performed using IBM SPSS Statistics for Windows, Version 23.0. Armonk, NY, USA: IBM Corp.

## 3. Results

Between 11 May 2020 and 14 June 2021, 1122 individuals were included in this research, of whom 741 were finally analysed. There were three inclusion periods (Figure 4).

### 3.1. Excluded Samples

A total of 145 patients sampled during the first inclusion period (11 May to 15 May 2020) were excluded. Their samples were analysed on the LoopDeelab device at an external private testing site in prefabricated premises. Following a report from the samplers concerning the rather low temperature at this testing site, a verification of sampling conditions was carried out after the first unblinding. Analysis of these conditions showed an on-site temperature below 15 °C (average temperature was 11 °C), which does not comply with the conditions of use of LoopDeelab according to manufacturer’s recommendations. The results obtained from patients enrolled at this testing site cannot therefore be used.

A total of 243 patients sampled during the second inclusion period (18 November to 12 December 2020) were also excluded. Following the second unblinding, complementary assays were made on the paired comparative samples whose RT-PCR and RT-LAMP results were discordant. We found that for several paired comparative samples, the results were still discordant when using the same assay for both (data not shown). This could be explained by the non-homogeneity of the paired comparative samples.

To solve these problems, two substantial modifications to this protocol were made on 17 August 2020 and on 6 April 2021. The first allowed the opening of a second testing site: the COVID-19 screening centre of Caen University Hospital. The second allowed the modification of the sampling method (Figure 1 and Figure 4).

### 3.2. Patient Characteristics

The majority of patients included in the study were sent for pre-hospitalisation and/or pre-operative screening tests. Another large proportion of the patients included were healthcare personnel sampled as part of the investigation of clusters or as part of monitoring their positivity.

The median age was 36 (Interquartile range = 57–21) and 53.98% of the patients were female (sex ratio = 0.85). According to their body mass index (BMI), 201 (27.46%) patients were overweight (25 < BMI < 30) and 86 (11.75%) were obese (BMI > 30) (eight of whom were morbidly obese (BMI > 40)). There were 97 (13.09%) symptomatic patients, and the most frequently reported symptoms were cough, rhinitis and headache.

### 3.3. Diagnostic Accuracy

Over the 741 samples, there were 28 RT-PCR “positive” results and four RT-PCR “indeterminate” results; among them, 21 had a Ct RdRp value < 33 (Table 1).

We calculate the intrinsic (sensitivity and specificity) and extrinsic (NPV: Negative predictive value and PPV: Positive predictive value) performances of LoopDeetect COVID-19 to detect the RdRp gene.

Among the 32 RT-PCR “positive” or “indeterminate” samples, only 26 of them had a RdRp Ct value < 40 and where therefore used for calculation of overall sensitivity and specificity of LoopDeetect COVID-19. Using RT-PCR results as the gold standard, there were 69.23% (CI 95%: 48.21–85.67) and 100% (CI 95%: 99.58–100.00), respectively (Figure 5). The NPV and PPV were 96.46% (CI 95%: 94.86–97.68) and 100% (CI 95%: 84.67–100.00).

There were two RT-PCR “positive” samples for which only the E and N genes were amplified with Ct values < 40. In both cases, the result was returned to the patient by the biologist with a comment specifying the weak positivity of the sample. They were sampled from the patient 18 and 23 days, respectively, after their first positive PCR test was sent to the virology department of Caen University Hospital.

The four RT-PCR “indeterminate” samples were checked using another RT-PCR assay. Among them, there are two samples for which the finally reported results—after checking—were weak positives. The results of the two other samples were finally reported “negatives”.

The intrinsic performances of LoopDeetect COVID-19 seem to evolve according to the RT-PCR RdRp Ct values. The sensitivity and specificity were, respectively, 85.71% (CI 95%: 63.66–96.95) (Figure 5) and 100% (CI 95%: 99.58–100.00) when only considering patients with RT-PCR RdRp Ct values < 33. The positive and negative predictive values were 100.00% (CI 95%: 84.67–100.00720) and 99.59% (CI 95%: 98.79–99.91), respectively. The representative curve of the evolution of sensitivity according to RT-PCR RdRp Ct values shows a gradually decreasing sensitivity for higher RT-PCR RdRp Ct values up to the originally accepted threshold of 40 (Figure 5).

### 3.4. Asymptomatic vs. Symptomatic

The overall sensitivity—when only considering samples with RT-PCR RdRp Ct value < 40—went from 40.00% (CI 95%: 12.16–73.76) among asymptomatic patients to 77.78% (CI 95%: 52.36–93.59) among symptomatic patients (Figure 5). 80.00% of symptomatic patients that were positive had RT-PCR RdRp Ct values < 33, whereas only 41.67% of asymptomatic patients had RT-PCR RdRp Ct values < 33 (*p = 0.053*). The mean RT-PCR Ct for the RdRp gene target was 22.26 (13.8–33.6) for symptomatic patients versus 27.56 (17.3–37.5) for asymptomatic patients (*p = 0.061*). (Figure 6, Table 2)

## 4. Discussion

Rapid and accurate diagnosis of SARS-CoV-2 infection is essential for the management of the current COVID-19 outbreak. The present results show that the overall sensitivity and specificity of LoopDeetect COVID-19 are 69.23% [CI 95: 48.21–85.67] and 100% [CI 95: 99.58–100.00]. These are 85.71% [CI 95: 63.66–96.95] and 100% [CI 95: 99.58–100.00], respectively, when only considering RT-PCR positive nasopharyngeal samples with RdRp Ct values < 33. These results are in the range of several other studies that have evaluated the performance of different LAMP PCR assays without extraction, and which have shown that overall sensitivity varies from 48.0% to 71.7% [11,12,13,14].

Our gold standard method (Allplex 2019 n-Cov assay kit, Seegene, Korea) is a reliable assay whose analytical performance has been evaluated in several studies, showing sensitivities ranging from 98.2 to 100.0% and specificities ranging from 99.0 to 100.0%, according to the gold standard used in these studies [15,16]. Its performance has been rated as good or better than most commercial kits available, which make it an excellent gold-standard choice [17,18].

This RT-PCR assay amplified three gene targets (RdRp, N and E), two of which were specific to SARS-CoV-2 (RdRp and N). RdRp being the common target of both methods compared, we calculated the intrinsic performance of LoopDeetect COVID-19 to detect the RdRp gene. Among all the RT-PCR “positive” or “indeterminate” samples, only 26 of them had a RdRp Ct value < 40 and were therefore used for calculation of overall sensitivity and specificity. The six samples without amplification of the RdRp gene had Ct E and/or N gene values < 40 and were finally reported as “negative” or “weak positive”. Detection of E and/or N genes without RdRp could have three different origins. It could be explained by patients in the early or in the final stage of their infection, with thus a small quantity of viral RNA in their samples. RNA originates from infected human cells and the cycle of replication of SARS-CoV-2 within these cells includes generation of subgenomic (sg)RNAs. No sgRNA is produced for the RdRp gene, while sgRNAs for the N gene are the most abundant [19]. This is one of the reasons why the RdRp gene is less frequently detected than the N gene, especially in samples with a small quantity of viral RNA. It could also be explained by recovered patients with degradation products of viral RNA in their samples, or by a non-optimal sampling.

Of the 741 samples analysed with an Allplex 2019 n-Cov assay kit (Seegene, Korea), 32 were “positives” or “indeterminates” (25 for all targets, three for two targets and four for only one target) and 709 were “negatives” for all targets. According to the manufacturer’s recommendations, a sample is considered positive if at least two genes among the three (RdRp, N and E) were amplified with a Ct value under 40. However, the French Society for Microbiology (SFM) has advised to consider samples with three targets and Ct values under 23 as “strong positive”, and samples with Ct values > 33 as “weak positive” [20]. Moreover, the sensitivity of LoopDeetect COVID-19 was 85.71% [CI 95: 63.66–96.95] when only considering samples with RdRp Ct values < 33. We also observed a gradual decrease in sensitivity for higher RdRp Ct values up to the originally accepted threshold of 40, which is consistent with findings from other widely marketed RT-LAMP PCR kits [11,12,13,14]. A study about the assessment of the RT-LAMP PCR assay ID NOW^TM^ COVID-19 (Abbott, Chicago, IL, USA) using samples in VTM demonstrated that “the assay performs well for strong and moderately positive samples but has a dramatic reduction in sensitivity for weakly positive samples (Ct > 34)” [14].

However, it is essential to remember that the determination of Ct cut-off values for the interpretation of PCR has serious limitations. The Ct value is dependent on many factors, most of which are not clearly identified. Although the effects associated with the number of days since symptom onset or the targeted gene of the RT-PCR assay used have been reported, care should be taken when interpreting Ct values because of many other technical and biological issues. [21]

RdRp Ct values seem to be higher in the asymptomatic patient group. The decrease of sensitivity for RdRp Ct > 33 suggests a lower sensitivity for asymptomatic patients. However, no significant difference of sensitivity was found between asymptomatic and symptomatic patient groups.

The main limitation of this study is the low number of positives samples—especially samples with Ct RdRp > 33—which hampers the robustness of statistical analysis and does not allow us to highlight significant differences between asymptomatic and symptomatic patients. It is explained by the low incidence rate during the third inclusion period. Between April and June 2021, the mean COVID-19 incidence rate in Calvados was 151 per 100,000 inhabitants [22]. We also had to exclude 378 patients enrolled during the first and second inclusion periods due to heterogeneity of the paired samples or to the disregard of LoopDeelab’s conditions of use. It could also be explained by the type of screening center involved in this protocol. Indeed, this center (COVID-19 screening center of Caen University Hospital) mainly received patients referred for pre-hospitalisation and/or pre-operative screening.

In conclusion, the LoopDeetect COVID-19 method showed good performances for the detection of SARS-CoV-2 on nasopharyngeal swabs on an outpatient basis. Its intrinsic performances are not equivalent to those of RT-PCR tests with a sensitivity of 69.23% and a specificity of 100.00%, but they are equivalent to that of other commercial RT-LAMP PCR COVID-19 kits. The sensitivity reaches 85.71% and with a specificity of 10,000% for samples with RdRp Ct < 33.

To the best of our knowledge, LoopDeelab is the only LAMP PCR diagnostic device allowing such a fast and reliable analysis with low-cost equipment. This is a new and innovative technology, designed for field use. It should be used in case of limited access to laboratory infrastructure, especially during epidemics with a high attack rate. Its use in a laboratory can be coupled with a separate extraction system to improve its sensitivity, particularly in the case of mass screening. LoopDeelab has recently undergone improvements allowing, in particular, to connect it via Bluetooth to smartphones or to control it via a Windows application. To ensure a more reliable verdict, LoopDeeScience has since put in place an internal control. In addition, they are making it easier for laboratories to perform an extraction and are innovating on a small, portable and fast automatic extractor.

The development of other LoopDeelab detection kits could be useful for the management of epidemics with a high attack rate (particularly measles epidemics). Indeed, this device being portable facilitates the rapid application of health measures (closure of establishments, vaccinations, treatments, etc.).

## Figures and Tables

**Figure 1 viruses-14-02062-f001:**
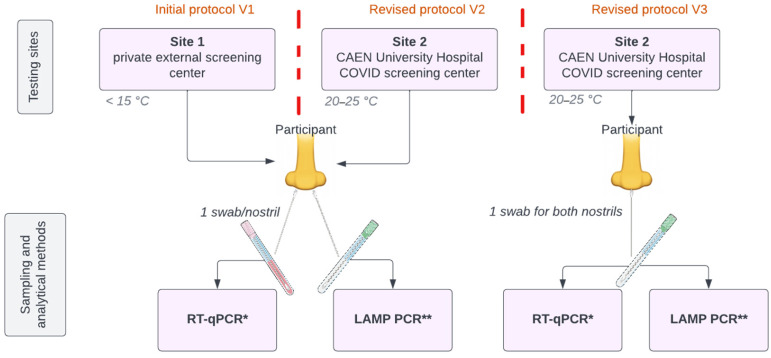
Sampling and testing site modifications. The first line shows the different testing sites and temperature conditions. The second line shows the different sampling and analytical methods. * RT-qPCR: Allplex 2019 n-Cov assay kit, Seegene, Korea. ** LAMP PCR: LoopDeetect COVID-19, LoopDeescience, Caen, France.

**Figure 2 viruses-14-02062-f002:**
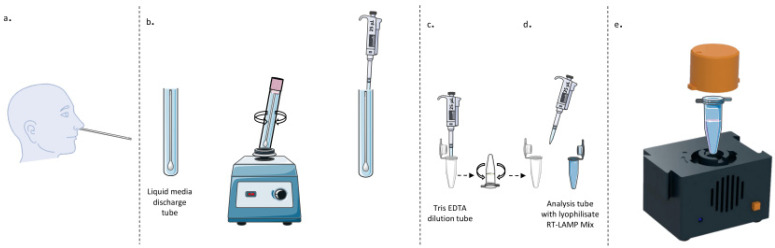
RT-LAMP assay (LoopDeetect COVID-19). After blowing the patient’s nose, the nasopharyngeal swab was performed with a flocked swab (**a**) and then placed in a liquid medium. After homogenisation, 25 µL of the liquid medium (**b**) were added to 475 µL of dilution buffer containing Tris EDTA (**c**). After homogenisation, 25 µL were added to a 1.5 mL microtube containing a lyophilised mix, including Master mix RT-LAMP and DNA-primer mix (**d**). The reaction tube carrying the sample was directly inserted into the LoopDeelab chamber (**e**). The scan was started by pressing the button on the device. The result was given directly without opening the tube, thus preventing contamination. The LoopDeelab (Loopdeescience, Caen, France) device concentrates technologies allowing us to heat the reaction tube and read different wavelengths after excitation by LED. A specific programme stored within the motherboard microprocessor controls the temperature and the wavelengths’ emission and reading (using diodes and spectrophotometer). An artificial intelligence calculates the figure of merit and gives a false or true result displayed directly on the LoopDeelab (red and green lights for positive and negative results, respectively).

**Figure 3 viruses-14-02062-f003:**
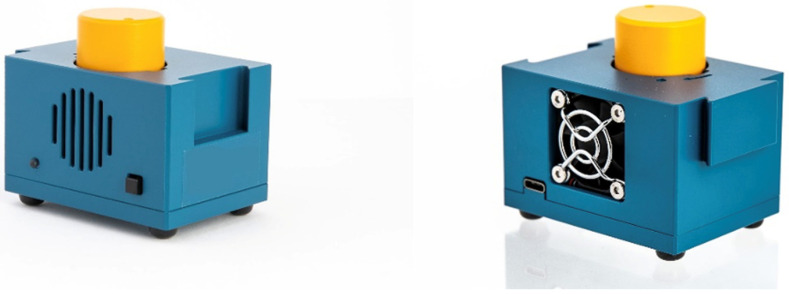
LoopDeelab device.

**Figure 4 viruses-14-02062-f004:**
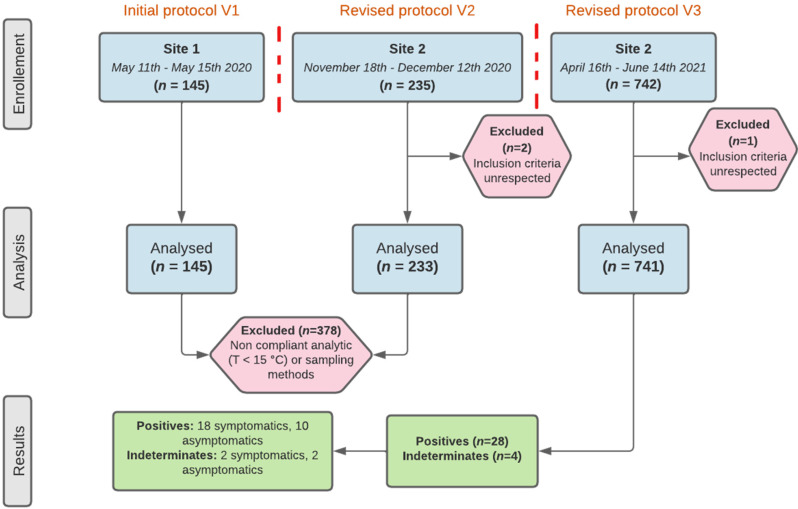
RAPID COVID study results flow chart. The first line of the flow chart shows the number of patients enrolled during the three inclusion periods. The second line shows the number of patients that were included in the analysis and the reasons why some were excluded. The third line shows the number of “positive” or “indeterminate” samples and the proportion of symptomatic patients. The red vertical dotted lines indicate protocol changes.

**Figure 5 viruses-14-02062-f005:**
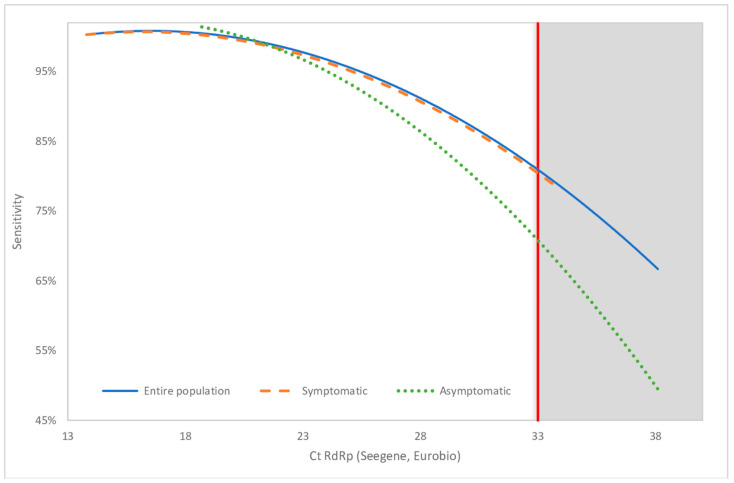
Evolution of sensitivity of LoopDeetect COVID-19 according to Ct RdRp values. Allplex 2019 n-Cov assay kit (Seegene, Korea) was the gold standard for calculation of sensitivity. The blue, red and green curves represent the entire, symptomatic and asymptomatic populations, respectively. The red vertical line indicates the threshold Ct = 33.

**Figure 6 viruses-14-02062-f006:**
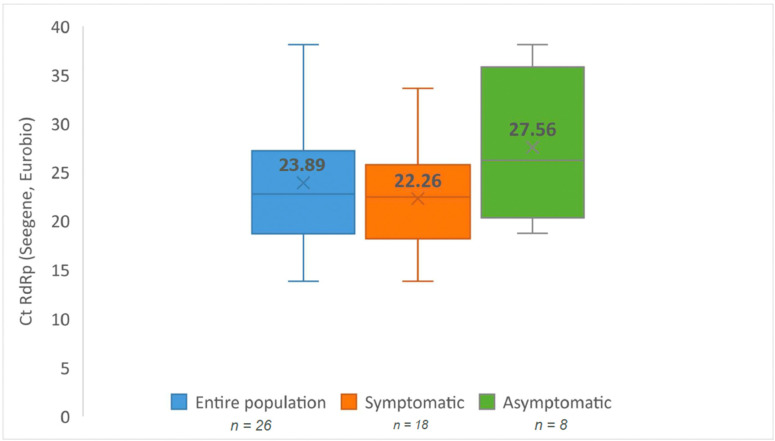
Box plots of Ct RdRp distributions in different groups. Ct RdRp distributions in the entire population (**left**), symptomatic group (**middle**) and asymptomatic group (**right**).

**Table 1 viruses-14-02062-t001:** RT-PCR and RT-LAMP results for different definitions of positivity.

	RT-PCR Positive	RT-PCR Indeterminate	RT-PCR Negative	Total
	** *Any gene with Ct ≤ 40 (RT-PCR), any patient* **			
**LAMP positive**	18	0	0	** * 18 * **
**LAMP negative**	10	4	709	** * 723 * **
** Total **	** * 28 * **	** * 4 * **	** * 709 * **	** * 741 * **
	** *Ct RdRp ≤ 33 (RT-PCR), any patient* **			
**LAMP positive**	18	0	0	** * 18 * **
**LAMP negative**	3	0	720	** * 723 * **
** Total **	** * 21 * **	** * 0 * **	** * 720 * **	** * 741 * **
	** *Any gene with Ct ≤ 40 (RT-PCR), symptomatic patients* **			
**LAMP positive**	14	0	0	** * 14 * **
**LAMP negative**	4	2	77	** * 83 * **
** Total **	** * 18 * **	** * 2 * **	** * 77 * **	** * 97 * **
	** *Ct RdRp ≤ 33 (RT-PCR), symptomatic patients* **			
**LAMP positive**	14	0	0	** * 14 * **
**LAMP negative**	2	0	81	** * 83 * **
** Total **	** * 16 * **	** * 0 * **	** * 81 * **	** * 97 * **
	** *Any gene with Ct ≤ 40 (RT-PCR), asymptomatic patients* **			
**LAMP positive**	4	0	0	** * 4 * **
**LAMP negative**	6	2	632	** * 640 * **
** Total **	** * 10 * **	** * 2 * **	** * 632 * **	** * 644 * **
	** *Ct RdRp ≤ 33 (RT-PCR), asymptomatic patients* **			
**LAMP positive**	4	0	0	** * 4 * **
**LAMP negative**	1	0	639	** * 640 * **
** Total **	** * 5 * **	** * 0 * **	** * 639 * **	** * 644 * **

**Table 2 viruses-14-02062-t002:** RdRp Ct values and diagnostic accuracy of RT-LAMP considering different patient groups.

	Entire Population	Symptomatic Patients	Asymptomatic Patients	*p*-Value
**Total** (n, %)	741 (100.00%)	97 (13.09%)	644 (86.91%)	//
**Positive and indeterminate samples** n (%)	32 (4.32%)	20 (20.62%)	12 (1.86%)	//
**Ct RdRp *mean ± sd***	23.89 ± 6.71	22.26 ± 5.58	27.56 ± 7.93	0.061 *
**Ct RdRp values** < 33 n (%)	21 (65.63%)	16 (80.00%)	5 (41.67%)	**0.053 ****
**RT-LAMP sensitivity (Ct RdRp < 40)** %, [CI 95]	69.23% [48.21–85.67]	77.78% [52.36–93.59]	40.00% [12.16–73.76]	0.097 **
**RT-LAMP sensitivity (Ct RdRp < 33)** %, [CI 95]	85.71% [63.66–96.95]	87.5% [61.65–98.45]	80.00% [23.86–99.49]	1.000 **
**RT-LAMP specificity (Ct RdRp < 40**) %, [CI 95]	100.00% [99.58–100.00]	100% [96.18–100.00]	100% [99.53–100.00]	//

* Student’s *t* test. ** Fisher’s exact test.

## Data Availability

Not applicable.

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
