# Peer review of "Use of LoopDeelab during the COVID-19 Pandemic: An Innovative Device for Field Diagnosis"

_viruses, 2022, doi:10.3390/v14092062_

Round 1
Reviewer 1 Report
This paper describes a validation of a novel RT-LAMP rapid assay for detection of SARS-CoV-2 virus using a small portable analyser.
The results of a double blind research study are presented comparing the novel RT-LAMP assay to an established RT-PCR “gold standard” assay.
The results in the paper are well presented and the figures are clear, however some minor modifications to the English and style of the writing are recommended.
On line 33, WHO should be defined, i.e. World Health Organization (WHO)
On Line 36, include a reference for the data
On Line 56, FAO should be defined, Food and Agricultural Organization?
On Line 154/155 I think 1.5ul microtube should be 1.5ml microtube?
Author Response
Please see the revised manuscript, we have revised according to the Reviewer #2
Reviewer 2 Report
In this article, Dossou et al. report the results of a double-blind controlled trial assessing a LAMP-based COVID-19 test named LoopDeelab. Overall, the design of trial is satisfactory. As explained hereafter, some modifications of the manuscript will render the results more intelligible to the readers. Provided that the authors include these corrections, the manuscript will be suitable for publication.
1) In the abstract, the authors only mention the results of the test for Ct < 33 (RT-PCR), i.e. “strong positive” patient sample containing a lot of viral genetic material. This corresponds to a specificity of 85.7% for the samples with Ct < 33, compared to a specificity of 69.2% for all samples. It is somewhat misleading for the reader to present the results this way. Results for all samples should be included in the abstract instead.
2) In the discussion (lines 356-360), the authors again bias their analysis towards samples with Ct < 33. This is not realistic considering that, in practice, the test needs to be able to pick up low positives, especially in a hospital context. In that regard, a specificity of 69.2% is quite low compared to RT-PCR tests. The authors should discuss more specifically the potential use of LoopDeelab and its limitation, explaining for example in which situation(s) it could be used/not used.
3) One main bias in diagnostic studies is patient selection. The authors briefly mention it (lines 230-235), but this should appear at the beginning of the results section instead, in greater details (what kind of patients? Hospitalized? COVID suspicion? Etc…)
4) The authors should cite previous work on RT-LAMP tests for SARS-CoV-2 in the introduction, not only in the discussion. It would also be interesting to write a brief paragraph comparing their results to other RT-LAMP tests.
5) The last result section (3.4 Symptomatic vs Asymtomatic) does not belong to the trial results, and should be removed from the manuscript. It only brings confusion to the reader.
6) The authors should use neutral pronouns when referring to patients (examples: “he” lines 100 and 110; “his” lines 83 and 151). Please replace throughout the manuscript:
- “he” by “he/she” or “they”
- “his” by “his/her” or “their”
